

# Bioinformatics analysis of microarray data to identify the candidate biomarkers of lung adenocarcinoma

Tingting Guo*, Hongtao Ma* and Yubai Zhou

Department of Biotechnology, College of Life Science & Bioengineering, Beijing University of Technology, Beijing, China
* These authors contributed equally to this work.

## ABSTRACT

**Background:** Lung adenocarcinoma (LUAD) is the major subtype of lung cancer and the most lethal malignant disease worldwide. However, the molecular mechanisms underlying LUAD are not fully understood.

**Methods:** Four datasets (GSE118370, GSE85841, GSE43458 and GSE32863) were obtained from the gene expression omnibus (GEO). Identification of differentially expressed genes (DEGs) and functional enrichment analysis were performed using the limma and clusterProfiler packages, respectively. A protein–protein interaction (PPI) network was constructed via Search Tool for the Retrieval of Interacting Genes (STRING) database, and the module analysis was performed by Cytoscape. Then, overall survival analysis was performed using the Kaplan–Meier curve, and prognostic candidate biomarkers were further analyzed using the Oncomine database.

**Results:** Totally, 349 DEGs were identified, including 275 downregulated and 74 upregulated genes which were significantly enriched in the biological process of extracellular structure organization, leukocyte migration and response to peptide. The mainly enriched pathways were complement and coagulation cascades, malaria and prion diseases. By extracting key modules from the PPI network, 11 hub genes were screened out. Survival analysis showed that except VSIG4, other hub genes may be involved in the development of LUAD, in which MYH10, METTL7A, FCER1G and TMOD1 have not been reported previously to correlated with LUAD. Briefly, novel hub genes identified in this study will help to deepen our understanding of the molecular mechanisms of LUAD carcinogenesis and progression, and to discover candidate targets for early detection and treatment of LUAD.

# INTRODUCTION

Lung cancer remains the one of leading healthy issues worldwide, with as estimated 2.1 million new cases and 1.8 million deaths in 2018 (*Bray et al., 2018*). It has been ranked the first and second cancer morbidity of male and female in China, respectively, and has the highest mortality rate (*Sun et al., 2018*). Lung adenocarcinoma (LUAD) is the most common subtype of lung cancer (*Maemura et al., 2018*; *Walters et al., 2013*). More than 60%

Corresponding author
Yubai Zhou, zhouyubai@bjut.edu.cn

of LUAD patients were observed harboring targetable gene alterations, which leading to remarkable responses in treating with tyrosine kinase inhibitors, and associated with improved survival rate (*Kris et al., 2014*). Despite the substantial advance in combined therapies, the prognosis of LUAD is still dismal, the 5-year survival rate is not over 20% (*Chen et al., 2014b*; *Ettinger et al., 2013*). Lacking sensitive and specific early biomarkers, a high possibility of drug resistance and metastasis is considered to contribute the high mortality of this disease. Therefore, there has a pressing need for identifying the more sensitive and specific biomarkers or drug targets of LUAD for developing effective diagnosis and treatment strategies.

Microarray technology provides an all-in-one system biology solution from hardware to software systems. It can simultaneously scan the hybridization signals of tens of thousands of gene probes in the chip and carry out quantitative analysis on the transcriptome profile of samples. Recent advances especially in the algorithms of probe signal detection and analysis, such as the introduction of artificial intelligence technologies, will make the results of microarray more accurate and reliable (*Gan et al., 2019a*, *2019b*; *Peng, 2006*). The microarray technique also provides a powerful tool for exploring the gene regulation pattern and molecular mechanisms involved in oncogenesis and progression of LUAD. Recently, different types of biomarkers including coding genes, miRNAs, long non-coding RNAs and circRNAs have been identified in lung cancer. Dysregulation of these molecules is involved in the tumor progression or is associated with the prognosis of patients (*Di et al., 2019*; *Vargas & Harris, 2016*; *Vencken, Greene & McKiernan, 2015*; *Wei & Zhou, 2016*). In view of the complexity of the molecular regulatory network of LUAD, current studies on tumor biomarkers are not sufficient. Therefore, it is still necessary to identify novel prognostic biomarkers, which will help us develop more sensitive and effective diagnostic and therapeutic strategies. However, limited sample size and significant variability among different projects make it hard to obtain credible results. In this study, four microarray datasets containing mRNA expression data between LUAD and non-cancerous tissues were downloaded from Gene Expression Omnibus (GEO) and the differentially expressed genes (DEGs) were screened out. Gene Ontology (GO), Kyoto Encyclopedia of Genes and Genomes (KEGG) and protein–protein interaction (PPI) network analyses were performed to explore the key modules and hub genes involved in LUAD progression. In sum, 349 DEGs and 10 hub genes were screened out, which may be candidate biomarkers for LUAD.

## MATERIALS AND METHODS

### Data download and pre-processing

Four datasets (GSE118370, GSE32863, GSE85841 and GSE43458) which contain the gene expression data of LUAD and normal tissues, were downloaded from GEO (https://www.ncbi.nlm.nih.gov/geo/) by getGEO function in package GEOquery (*Davis & Meltzer, 2007*). The detail information of GEO datasets was listed in Table 1. The raw expression files of four microarray datasets were pre-processed according to the method described previously with minor modifications (*Giulietti et al., 2016*). Briefly, the CEL format files were input and background correction and normalization were

**Table 1 The detail information of four GEO datasets.**

| ID | Tissue | Platform | Normal | Tumor |
|---|---|---|---|---|
| GSE118370 | LUAD | GPL570 | 6 | 6 |
| GSE85841 | LUAD | GPL20115 | 8 | 8 |
| GSE43458 | LUAD | GPL6244 | 30 (never-smoker) | 40 (never-smoker) |
| GSE32863 | LUAD | GPL6884 | 58 | 58 |

**Note:**
GEO, Gene Expression Omnibus; LUAD, lung adenocarcinoma.

conducted using the Robust multichip average function implemented in affy package in R environment (*Bolstad et al., 2003*; *Irizarry et al., 2003*). Next, the array probes were converted into matched gene symbols according to annotation information. In case of multiple probes corresponding to a single gene, the value of gene expression was designated as the mean of the probes. Then, the batch effects among different platforms were removed by ComBat function of the sva package (*Leek et al., 2012*). Finally, the normalized microarray-based data of four datasets were merged into a single global dataset which contained a total of 12,926 common genes in all four GEO datasets.

## Identification of DEGs

Identifying DEGs in different disease states and investigating their functions and interactions may help to unravel potential regulatory mechanisms for disease occurrence and progression. In the present study, the DEGs between LUAD and normal tissues were identified by limma package (*Ritchie et al., 2015*). The Benjamini–Hochberg procedure was introduced to reduce the false positive rate (FDR) in multiple comparisons (*Benjamini & Hochberg, 1995*). Genes with |log2 Fold Change| $\geq$ 1 and FDR < 0.05 were considered as DEGs.

## GO and KEGG enrichment analysis

Gene Ontology and KEGG analyses were conducted using enrichGO and enrichKEGG functions of clusterProfiler package, respectively (*Yu et al., 2012*). p.adjust (FDR) < 0.05 was considered to be statistically significant.

## Construction of PPI network and module analysis

The network of proteins interaction provides valuable clues for understanding the molecular mechanisms underlying the progress of carcinoma. The PPI network was constructed by Search Tool for the Retrieval of Interacting Genes (STRING) (https://string-db.org/) with interaction score of 0.9 as the threshold. Subsequently, the candidate modules were detected by Cytoscape plugin molecular complex detection (MCODE) with default parameters: degree cut-off = 2, node score cut-off = 0.2, $k$-core = 2, and max depth = 100.

## Hub gene analysis

The seed genes in modules referred to hub genes. The overall survival analyses were performed using online tool Kaplan–Meier Plotter (http://kmplot.com/) (*Gyorffy et al., 2013*). The logrank *P* < 0.05 was considered statistically significant. The association of
**Table 2 Top ten up- and down-regulated DEGs.**

| Up-regulated DEGs | | | Down-regulated DEGs | | |
| --- | --- | --- | --- | --- | --- |
| Gene symbol | Log2FC | FDR | Gene symbol | Log2FC | FDR |
| SPP1 | 3.08474 | 1.74E-33 | FABP4 | −3.37555 | 1.25E-47 |
| OCIAD2 | 1.477746 | 1.13E-29 | STX11 | −2.01766 | 1.51E-44 |
| ETV4 | 1.366623 | 5.38E-27 | CAV1 | −2.70539 | 6.04E-43 |
| TOP2A | 1.777989 | 3.24E-26 | FHL1 | −2.3099 | 5.77E-42 |
| COL10A1 | 1.364488 | 3.74E-26 | TEK | −2.30698 | 1.09E-40 |
| PROM2 | 1.570558 | 7.77E-26 | AGER | −2.69928 | 6.05E-40 |
| MMP11 | 1.929847 | 1.03E-23 | FMO2 | −2.44791 | 1.42E-39 |
| UBE2T | 1.154061 | 5.78E-23 | CRYAB | −1.98382 | 1.77E-39 |
| ABCC3 | 1.444218 | 3.71E-22 | GRK5 | −1.4854 | 1.53E-38 |
| BAIAP2L1 | 1.067336 | 4.15E-22 | TMEM100 | −2.928 | 4.79E-38 |

**Note:**
  DEGs, differentially expressed genes; FDR, false discovery rate; Log2FC, Log2 (Fold Change).

expression level of hub genes with clinical traits were analysis using the Oncomine database (*Rhodes et al., 2007*).

# RESULTS

## Data preprocessing and DEG screening

Four GEO datasets were downloaded, pre-processed and merged into a global dataset which contained 112 LUAD and 102 normal samples. Totally, 349 DEGs were identified by limma package (*Ritchie et al., 2015*), including 74 up-regulated genes and 275 down-regulated genes. The most statistically significant up-regulated and down-regulated genes are listed in Table 2. The distribution of DEGs was presented by volcano plot (Fig. 1).

## GO and KEGG analysis

The biological functions and pathways analyses were conducted using R package clusterProfiler (*Yu et al., 2012*). The GO categories of biological process (BP), cellular component (CC) and molecular function (MF) were enriched respectively (Figs. 2A–2C) and the top 15 GO terms of the up-regulated and down-regulated DEGs were listed in Tables S1 and S2, respectively. The up-regulated DEGs were mainly associated with extracellular matrix (ECM) processing, such as ECM organization (BP, GO:0030198), ECM (CC, GO:0031012) and serine-type endopeptidase activity (MF, GO:0004252). The down-regulated DEGs were most significantly related to response to corticosteroid (BP, GO:0031960), ECM (CC, GO:0031012) and growth factor binding (MF, GO:0019838). The DEGs were mainly enriched in pathways of complement and coagulation cascades (hsa04610), malaria (hsa05144), prion diseases (hsa05020), fluid shear stress and atherosclerosis (hsa05418), AGE-RAGE signaling pathway in diabetic complications (hsa04933), vascular smooth muscle contraction (hsa04270), IL-17 signaling pathway (hsa04657), leukocyte transendothelial migration (hsa04670), protein digestion and absorption (hsa04974) and drug metabolism—cytochrome P450 (hsa00982) (Table 3; Fig. 2D).

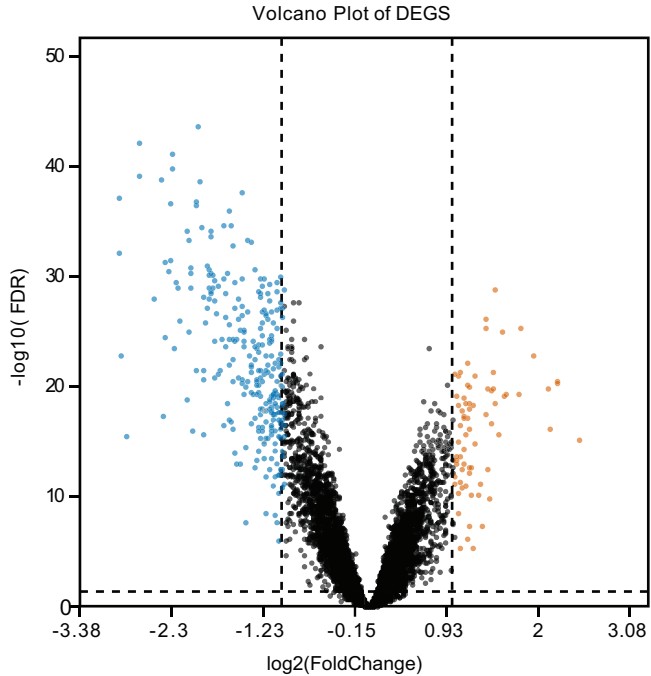

**Figure 1** **Volcano plot of the DEGs.** The vermilion and blue dots represent DEGs filtered based on the cut-off values of |log2FoldChange| > 1.0 and adjusted *P*-value (FDR) < 0.05, while the black dots represent genes that are not satisfied the cut-off values of differential expression. The horizon dotted line indicates the position of −log10 (FDR) = 0.05, and the vertical dotted lines indicate the positions of |log2FoldChange| = 1.0. Vermilion, upregulation; blue, downregulation. DEGs, differentially expressed genes; FDR, false discovery rate.

## Construction of PPI network and module analysis

Protein–protein interaction network reflect the spatiotemporal relationship of macromolecules within the cell which will provide valuable information about molecular mechanisms in physiological and pathological process. To explore the molecular mechanisms underlying LUAD progression, the online STRING database was applied to construct the PPI network. The interaction score of 0.9 (highest confidence) was set as threshold, and nodes without connections were removed from network. Finally, the PPI network consisted of 349 nodes with 277 edges, and average local clustering coefficient was 0.337 (PPI enrichment *P*-value < 1.0E-16) (Fig. 3A). Then, the key modules were identified via MCODE plugin. A total of 11 functional clusters of modules and related hub genes were detected. The top three significant modules were presented in Figs. 3C–3E. The KEGG analysis of module genes revealed that the top three modules were mainly associated with the chemokine signaling pathway (hsa04062), complement and coagulation cascades (hsa04610), human cytomegalovirus infection (hsa05163) and vascular smooth muscle contraction (hsa04270) (Fig. 3B).

## Hub genes analysis

A total of 11 genes were identified as hub genes. The overall survival analysis of the hub genes was performed using Kaplan–Meier curve. Except VSIG4, LUAD patients with downregulated ADAMTS8, AOX1, EFEMP1, METTL7A, MYH10, PTGER4, TMOD1,
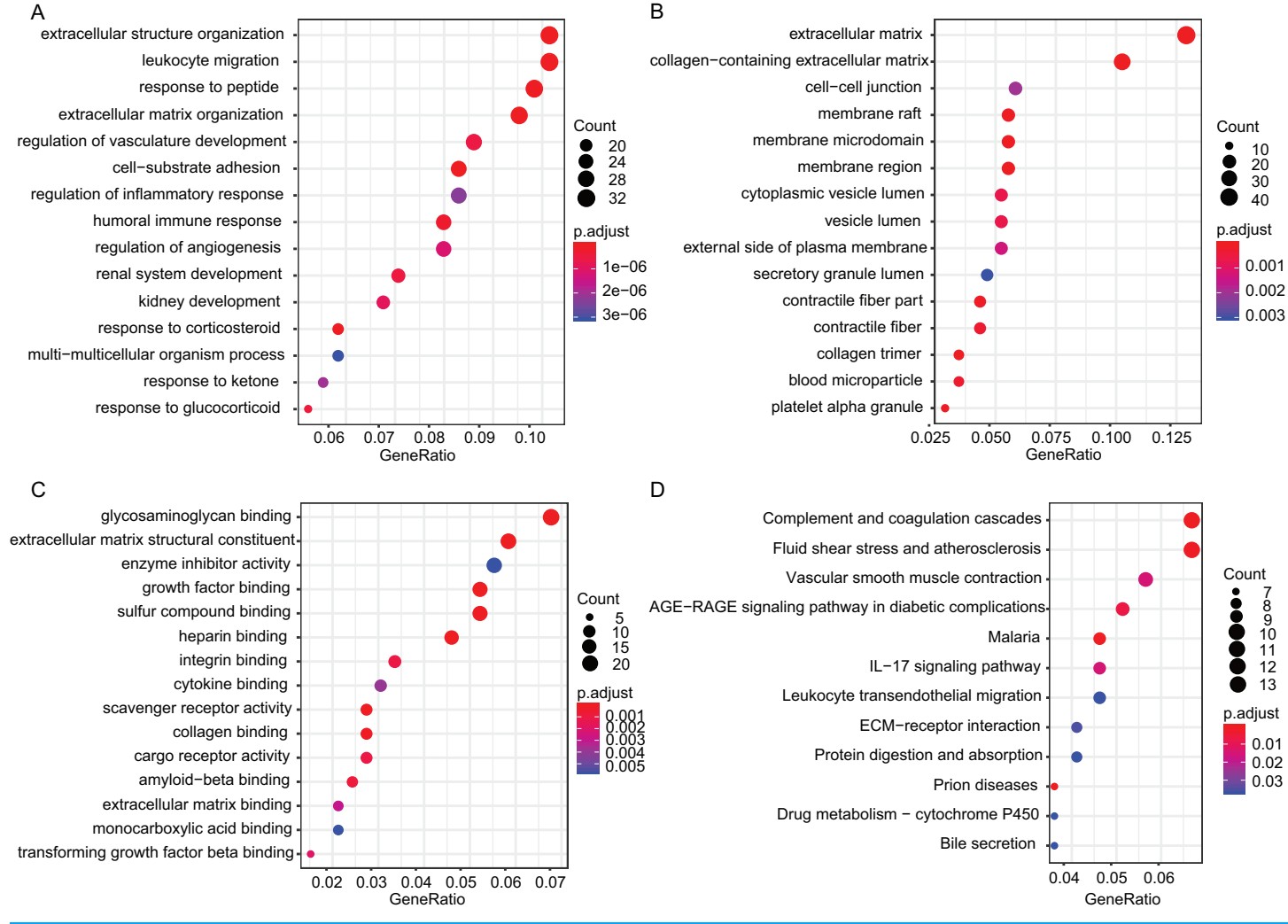

**Figure 2 GO and KEGG analysis of DEGs.** (A–C) The top 15 terms of GO categories of biological process (BP), cellular component (CC) and molecular function MF, respectively. (D) KEGG pathway analysis of DEGs, p.adjust (FDR) < 0.05 was considered significantly. GO, Gene Ontology; KEGG, Kyoto Encyclopedia of Genes and Genomes; FDR, false discovery rate.

CDH13 and upregulated PRC1 showed worse overall survival (Figs. 4A–4I). It is worth noting that FCER1G is downregulated in LUAD patients, but the low expression level is associated with better overall survival (HR = 1.87) (Fig. 4J). Subsequently, the expression status of hub genes with HR < 0.5 or HR > 2 were further validated using the Oncomine database. The result showed that ADAMTS8, METTL7A and MYH10 were significantly downregulated and PRC1 was markedly overexpressed in LUAD in the different datasets (Fig. 5). In the Okayama Lung dataset, the alternation of ADAMTS8, METTL7A, MYH10 and PRC1 were associated with tumor grade (Fig. 6), implicating vital roles of these genes in the carcinogenesis or progression of LUAD.

## DISCUSSION

In this study, four GEO datasets were analyzed and 349 DEGs were identified, including 74 up-regulated and 275 down-regulated genes. The KEGG analysis revealed the top three

**Table 3  KEGG enriched pathways of DEGs.**

| ID | Description | Count | p.adjust (FDR) |
|---|---|---|---|
| hsa04610 | Complement and coagulation cascades | 13 | 1.12E-05 |
| hsa05144 | Malaria | 9 | 0.000271 |
| hsa05020 | Prion diseases | 7 | 0.001406 |
| hsa05418 | Fluid shear stress and atherosclerosis | 13 | 0.001921 |
| hsa04933 | AGE-RAGE signaling pathway in diabetic complications | 10 | 0.007362 |
| hsa04270 | Vascular smooth muscle contraction | 11 | 0.014433 |
| hsa04657 | IL-17 signaling pathway | 9 | 0.014433 |
| hsa04974 | Protein digestion and absorption | 8 | 0.039299 |
| hsa04670 | Leukocyte transendothelial migration | 9 | 0.039299 |
| hsa00982 | Drug metabolism—cytochrome P450 | 7 | 0.039299 |

Note:
KEGG, Kyoto Encyclopedia of Genes and Genomes; DEGs, differentially expressed genes; FDR, false discovery rate.

enriched pathways were complement and coagulation cascades, malaria and prion diseases. These annotation results provided valuable clues to reveal molecular interactions in the development of LUAD. Indeed, the complement system has been reported to play critical roles in tumor progression (Zhao et al., 2019). The overexpressed C3 in blood and downregulated C3 in tumor tissues were observed in lung cancer patients, and related to poorer prognosis (Ajona et al., 2013; Lin et al., 2014; Mehan et al., 2012). The confusing experimental results suggest that the complement system may be involved in complex tumor regulatory processes by reshaping tumor microenvironment, which is worthy of further study. In recent years, malaria and prion disease, which had not previously attracted much attention, have also been found to be associated with tumors. Epidemiological study has shown that the incidence of malaria is negatively correlated with the mortality of colorectal cancer, breast cancer and lung cancer (Qin et al., 2017). The proposed anti-tumor mechanisms included systemly stimulating the immune responses (Chen et al., 2011) and inhibiting key pathways in tumor progress (Deng et al., 2018). Previously, prion protein (PrPc) was thought to play a role only in the central nervous system. However, accumulating evidence shows that PrPc has wider biological functions that were not previously expected (Mehrpour & Codogno, 2010). PrPc may be associated with the biology of many cancers, and overexpression of PrPc promotes the proliferation, invasion and metastasis of the gastric cancer cell (Liang et al., 2009; Mehrpour & Codogno, 2010; Pan et al., 2006). These data may provide new ideas and directions for the mechanism research and therapeutic strategy of LUAD. The GO analyses indicated that DEGs were significantly related to BP of ECM organization and process. Previous studies have reported that ECM remodeling promotes cancer progression and is associated with a poor prognosis in lung cancer patients (Kopparam et al., 2017; Xia et al., 2012). The up- and down-regulated DEGs were simultaneously enriched into ECM process, which is consistent with the propensity for metastasis and highly invasive characteristics of LUAD.

In line with the GO analysis, among 11 hub genes identified by PPI network and modules analysis, EFEMP1, PTGER4, ADAMTS8, CDH13, MYH10 and METTL7A are

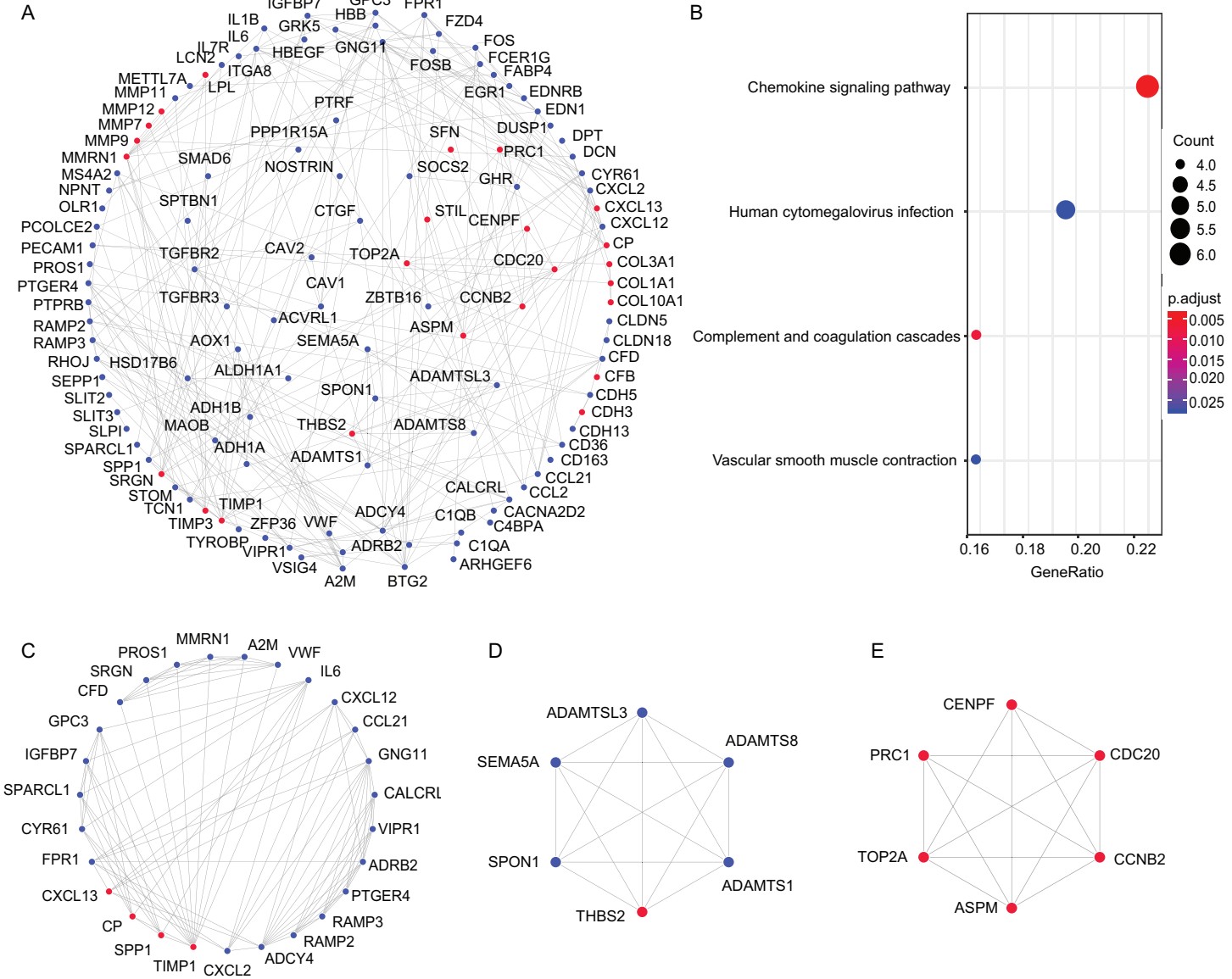

**Figure 3 Protein–protein interaction network of DEGs and modules analysis.** (A) PPI network of DEGs; (B) The KEGG enrichment analysis of the genes in the top three modules. (C–E) the top 3 modules of PPI network. The red nodes represent the upregulated DEGs. The blue nodes represent the downregulated DEGs. PPI, protein–protein interaction; DEG, differentially expressed gene; KEGG, Kyoto Encyclopedia of Genes and Genomes.

associated with ECM process, and survival analysis indicated that down-regulation of these hub genes was associated with worse overall survival. EFEMP1 plays distinct biological functions in different tumors. In osteosarcoma and gliomas, EFEMP1 is overexpressed and promotes the invasion and metastasis of tumor cells in vitro and in vivo by activating the expression of MMP-2 and notch signaling, respectively (*Hu et al., 2009*, *2012*; *Wang et al., 2015*), while in gastric cancer, endometrial carcinoma, hepatocellular carcinoma and lung cancer, EFEMP1 is downregulated, and is proposed as a prognosis biomarker (*Chen et al., 2014a*; *Kim et al., 2014*; *Nomoto et al., 2010*; *Yang et al., 2013*;

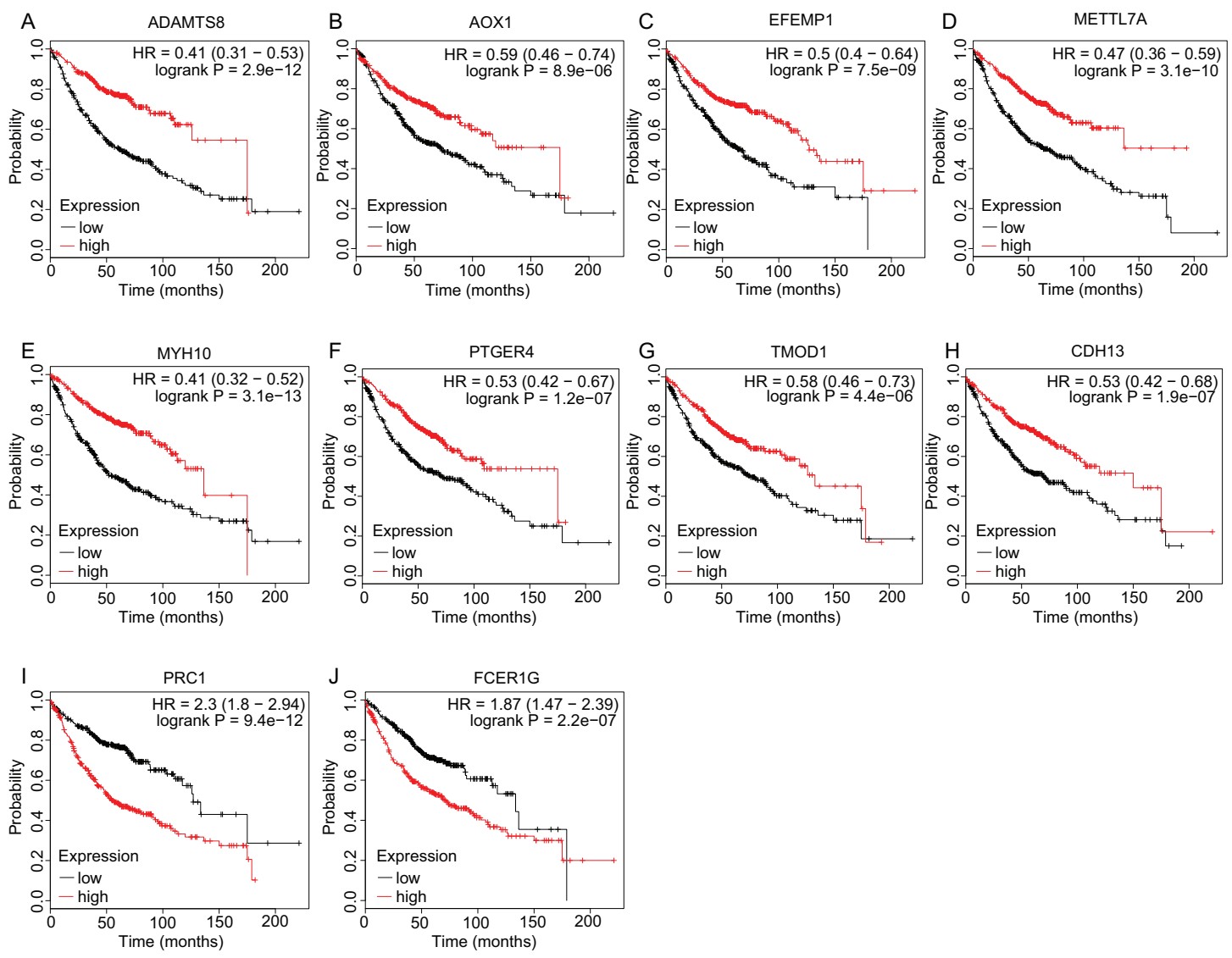

**Figure 4 Overall survival analyses of hub genes.** (A–J) The overall survival analyses of hub genes were performed using Kaplan–Meier Plotter online platform. Logrank *P* < 0.05 was considered statistically significant.

*Yue et al., 2007*; *Zhu et al., 2014*). Overexpression of EFEMP1 has been reported to suppressed invasion and migration of LUAD cells via inhibiting the epithelial-to-mesenchymal transition pathway (*Kim et al., 2014*). PTGER4 is overexpressed and proposed as a therapeutic target for LUAD and other cancers (*Doherty et al., 2009*; *Fulton, Ma & Kundu, 2006*; *Heinrichs et al., 2018*; *Kim et al., 2010*; *Ma et al., 2006*; *Xin et al., 2012*). The above report is inconsistent with our results, which may be due to the heterogeneity of the tumor and the limited number of samples. Therefore, subsequent large sample functional verification is required. ADAMTS8 encodes an inactive proenzyme and forms mature active enzyme by proteolysis (*Apte, 2009*). ADAMTS8 is identified as a secretory angiogenesis inhibitor that inhibits VEGF-mediated angiogenesis by blocking the EGFR signaling pathway (*Choi et al., 2014*; *Dunn et al., 2006*; *Vazquez et al., 1999*).

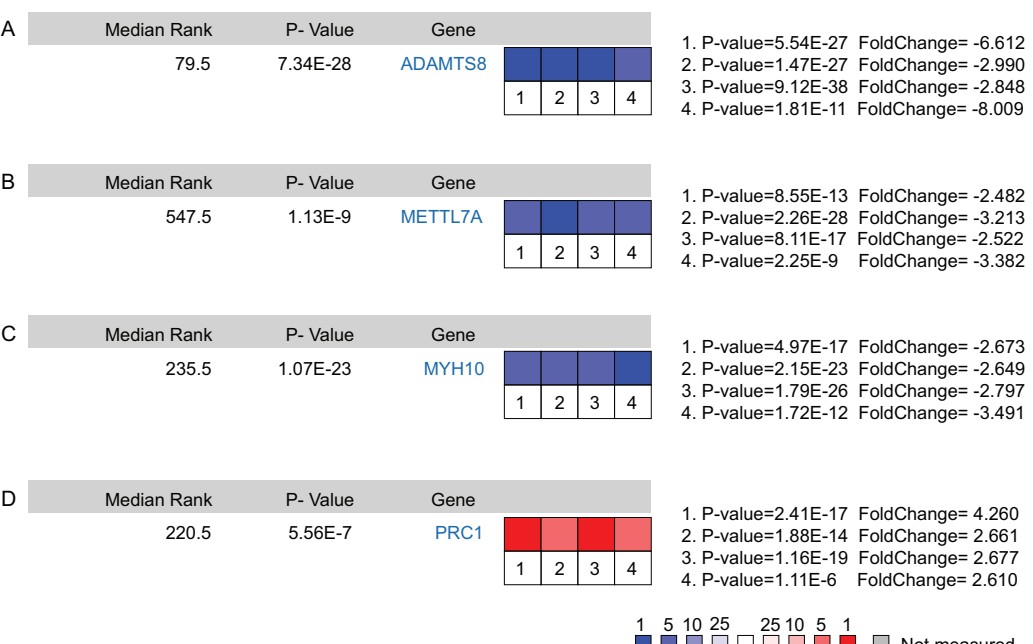

**Figure 5 Oncomine analysis of cancer vs. normal tissue of ADAMTS8, METTL7A, MYH10 and PRC1.** Heat maps of ADAMTS8 (A), METTL7A (B), MYH10 (C) and PRC1 (D) gene expression in lung adenocarcinoma samples vs. normal tissues. (1) Lung Adenocarcinoma vs. Normal, Hou Lung (*Hou et al., 2010*); (2) Lung Adenocarcinoma vs. Normal, Landi Lung (*Landi et al., 2008*); (3) Lung Adeno-carcinoma vs. Normal, Selamat Lung (*Selamat et al., 2012*); (4) Lung Adenocarcinoma vs. Normal, Su Lung (*Su et al., 2007*).

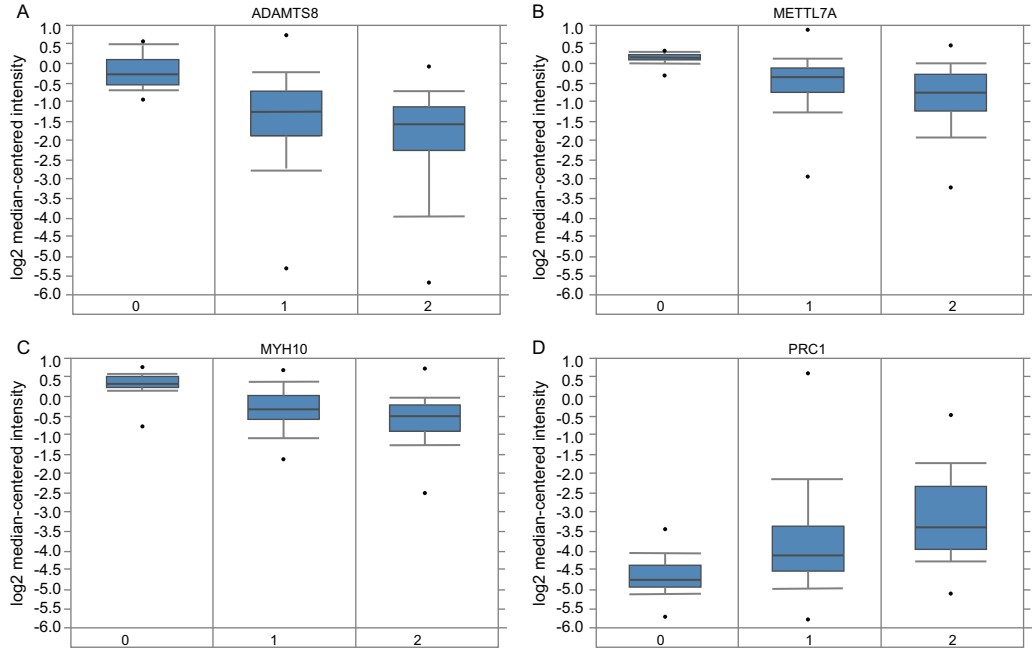

**Figure 6 The association between the expression of selected hub genes and tumor stage.** (A–D) The expressions of ADAMTS8, METTL7A, MYH10, PRC1 were correlated with tumor stage in the Okayama Lung dataset. 0: No value; 1: Stage I; 2: Stage II.

Downregulated ADAMTS8 in some cancers including NSCLC (*Heighway et al., 2002*; *Huang et al., 2019*; *Masui et al., 2001*; *Porter et al., 2004*; *Rodriguez-Rodero et al., 2013*; *Zhao et al., 2018*) is associated with poorer prognosis (*Drilon et al., 2014*; *Li et al., 2015*; *Porter et al., 2006*). CDH13 encodes a member of cadherin superfamily. The hypermethylation in promoter region of CDH13 was frequently observed in lung cancer, and proposed to correlate to drug sensitivity and poorer prognosis (*Kontic et al., 2012*; *Toyooka et al., 2006*; *Zhai & Li, 2014*; *Zhong et al., 2015*). As the only upregulated hub gene, PRC1 is involved in the process of cytokinesis and is upregulated in breast cancer, hepatocellular carcinoma, and lung cancer (*Jiang et al., 1998*; *Liu et al., 2018*; *Shimo et al., 2007*; *Zhan et al., 2017a*). Recent research has found that it promoted proliferation and metastasis of LUAD cells via potentiating the Wnt/β-catenin pathway (*Zhan et al., 2017b*), and inhibiting the expression of PRC1 in LUAD cells by miR-1-3p was reported to suppress tumorigenesis (*Li et al., 2019*).

Literature retrieval showed that the relation of LUAD and hub genes MYH10, METTL7A, FCER1G and TMOD1 has not been reported. MYH10 belongs to the myosin superfamily, which can regulate ECM remodeling (*Kim et al., 2018*; *Kim, Kurie & Ahn, 2015*). Methyltransferase METTL7A was found to facilitate Hepatitis C Virus propagation via recruitment of NS4B (*Park et al., 2015*). FCER1G encodes a high affinity IgE receptor which is associated with prognosis of renal clear cell carcinoma (*Chen et al., 2017*). As an aldehyde oxidase, TMOD1 is an actin-capping protein involved in regulation of the length and depolymerization of actin filaments. Overexpression of TMOD1 promotes cell proliferation of breast cancers and metastasis of oral squamous cell carcinoma (*Ito-Kureha et al., 2015*; *Suzuki et al., 2016*). However, the molecular mechanism of these newly identified hub genes in LUAD remains largely unknown, and further functional studies were warranted.

In present study, we have identified a set of candidate biomarkers that may play an important role in the progression of LUAD. These newly identified hub genes could be used as research subjects for exploring their roles in the disease process, so as to further deepen our understanding of the molecular mechanism of LUAD, and also as potential prognostic biomarkers for clinical validation studies to clarify their prognostic effects. However, there are still some limitations in this study. First, this study is based on bioinformatics analysis of published data and lacks experimental verification, and the study cannot determine whether there is a causal relationship between the differential expression of hub genes and disease progression. Finally, although we combined four GEO datasets, the number of samples is still relatively small, which may lead to potential unreliable results. Therefore, subsequent bioinformatics analysis and experimental verification with larger samples are necessary.

## CONCLUSIONS

In summary, this study identified several DEGs by integrating four GEO datasets and extracted 11 hub genes from PPI network, among which 10 hub genes were shown to be related to the occurrence and development of LUAD and four hub genes have not been previously reported but may play an important role in LUAD. The molecular mechanism

of these novel hub genes in LUAD is worthy of further study, and relevant prognostic model can also be constructed based on these genes for risk assessment, classification and prognostic judgment of patients with LUAD.

### Funding

This work was supported by the Major National Research and Development Projects (2012ZX10001005-007), the National High Technology Research Program of China (2012AA02A404), and the State Key Laboratory for Infectious Disease Prevention and Control (2011SKLID103). The funders had no role in study design, data collection and analysis, decision to publish, or preparation of the manuscript.

### Grant Disclosures

The following grant information was disclosed by the authors:
Major National Research and Development Projects: 2012ZX10001005-007.
National High Technology Research Program of China: 2012AA02A404.
State Key Laboratory for Infectious Disease Prevention and Control: 2011SKLID103.

### Competing Interests

The authors declare that they have no competing interests.

### Author Contributions

- Tingting Guo performed the experiments, analyzed the data, prepared figures and/or tables.
- Hongtao Ma performed the experiments, analyzed the data, prepared figures and/or tables.
- Yubai Zhou conceived and designed the experiments, performed the experiments, analyzed the data, contributed reagents/materials/analysis tools, prepared figures and/or tables, authored or reviewed drafts of the paper, approved the final draft.

### Data Availability

Data is available at NCBI GEO under accession numbers GSE118370, GSE32863, GSE85841 and GSE43458.

### Supplemental Information

Supplemental information for this article can be found online at http://dx.doi.org/10.7717/peerj.7313#supplemental-information.

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
