# Peer review of "Bioinformatics analysis of microarray data to identify the candidate biomarkers of lung adenocarcinoma"

_PeerJ, doi:10.7717/peerj.7313_

## Round 0.1 · original submission · Major Revisions

Please address the comments/feedback presented by the reviewers.

Reviewer 1 ·

Basic reporting

This paper is nice, and has real-world applications; I see many positive aspects in this work and would like to see it published. In fact, this paper will be of value and interest to as a significant portion of potential readers of the journal.

Experimental design

The experimental design and results seem to be much appreciable, while the authors might discuss and list the contributions of this paper.

Validity of the findings

This paper identified several differentially expressed genes by integrating four GEO 230 datasets and extracted 11 hub genes from PPI network.

Additional comments

This paper identified several differentially expressed genes by integrating four GEO 230 datasets and extracted 11 hub genes from PPI network. This paper is nice, and has mathematical real-world applications; I see many positive aspects in this work and would like to see it published. In fact, this paper will be of value and interest to as a significant portion of potential readers of the journal. In my opinion, this paper can be further improved in the following aspects:

1. The experimental results seem to be much appreciable, while the authors might discuss and list the contributions of this paper.

2. More detailed review of the literature is expected in separate section. Specially, it is required that the previous solutions to this problem be addressed. Then, the advantages (and disadvantages?) of the proposed work and should be discussed.

3. Some parts of mathematic derivations are not given in details. I suggest the authors a carefully checking and give all the necessary manipulations for the derived formulas.

4. The authors should clarify if the work requires data processing or not. More detail information should be added to make clear explanations.

5. Some latest references about Microarray and related methods should be added to give readers an up-to-date picture. In this sense, the following papers can be referred:
[a1]A novel ensemble machine learning for robust microarray data classification. Computers in Biology and Medicine. 2006 Jun 1;36(6):553-73.
[a2]Wavelet denoising algorithm based on NDOA compressed sensing for fluorescence image of Microarray, IEEE Access, vol. 7, no. 1, pp. 13338-13346, 2019.
[a3] Multilevel segmentation optimized by physical information for gridding of Microarray images, IEEE Access, vol. 7, no. 1, pp. 32146-32153, 2019.

6. Some future directions can be discussed in the conclusion part.

7. The authors still need a careful check of English, formulas and format/style.

I recommend the publication of this paper after the authors address the above concerns.

·

Basic reporting

Clear and unambiguous, professional English used throughout (X).

Experimental design

No comment.

Validity of the findings

Conclusions are well stated, linked to original research question & limited to supporting results (X).

Additional comments

In this study, through combining integrative bioinformatics analyses, Guo et al identified 11 hub genes, among which 10 genes were shown to be related to the occurrence and development of LUAD. Finally, the authors concluded that 6 hub genes might be regarded as potential biomarkers. The study is interesting, however, in my opinion, the manuscript is not of PeerJ standard in its current version.

Other remarks:
1. This is an integrated bioinformatics study based on the published data without any experimental validations, thus, the Title seems to be very ambiguous.
2. The Discussion is too long. There are lots of re-description of the results, such as the Third and Fourth paragraphs.
3. Line 213-214, the authors claim that the relation of LUAD and these identified 6 hub genes has not been reported. This description is inaccurate (Kontic et al., 2012; Stavrinou et al., 2015; Zhai and Li, 2014; Zhong et al., 2015).
4. The figures and tables do not meet the standards for publication, especially that the font and its size should be uniform and clear. The authors should carefully revise all these figures and tables.
5. If possible, the authors should use qRT-PCR or western blot to validate the expression of these selected hub genes, and it will be preferable to employ the gain- and loss of function studies in vitro.

References:
Kontic, M., Stojsic, J., Jovanovic, D., Bunjevacki, V., Ognjanovic, S., Kuriger, J., Puumala, S., and Nelson, H.H. (2012). Aberrant promoter methylation of CDH13 and MGMT genes is associated with clinicopathologic characteristics of primary non-small-cell lung carcinoma. Clinical lung cancer 13, 297-303.
Stavrinou, P., Mavrogiorgou, M.C., Polyzoidis, K., Kreft-Kerekes, V., Timmer, M., Marselos, M., and Pappas, P. (2015). Expression Profile of Genes Related to Drug Metabolism in Human Brain Tumors. PloS one 10, e0143285.
Zhai, X., and Li, S.J. (2014). Methylation of RASSF1A and CDH13 genes in individualized chemotherapy for patients with non-small cell lung cancer. Asian Pacific journal of cancer prevention : APJCP 15, 4925-4928.
Zhong, Y.H., Peng, H., Cheng, H.Z., and Wang, P. (2015). Quantitative assessment of the diagnostic role of CDH13 promoter methylation in lung cancer. Asian Pacific journal of cancer prevention : APJCP 16, 1139-1143.

---

## Round 0.2 · accepted · Accept

Please perform an additional proof reading on the final version.

Reviewer 1 ·

Basic reporting

Results shown that the proposed method is successful and efficient. Moreover, this paper is very well written and organized, and the contributions are well conveyed.

Experimental design

Results shown that the proposed method is successful and efficient.

Validity of the findings

This paper is very well written and organized, and the contributions are well conveyed.

Additional comments

The new version has been more clearly written and the results have been more rigorously proved. I think that the paper is suitable for publication.

·

Basic reporting

No comment.

Experimental design

No comment.

Validity of the findings

No comment.

Additional comments

No comment.